# ANTARES and KM3NeT: The Latest Results of the Neutrino Telescopes in the Mediterranean†

**Matteo Sanguineti** [1,2]* on behalf of ANTARES and KM3NeT collaborations

1    Dipartimento di Fisica dell'Università, Via Dodecaneso 33, 16146 Genova, Italy
2    INFN - Sezione di Genova, Via Dodecaneso 33, 16146 Genova, Italy
*    Correspondence: matteo.sanguineti@ge.infn.it
†    This Paper Is Based on the Talk at the 7th International Conference on New Frontiers in Physics (ICNFP 2018), Crete, Greece, 4–12 July 2018.

**Abstract:** The measurement of cosmic neutrinos is a new and unique method to observe the Universe. Neutrinos are chargeless, weakly-interacting particles that can provide information about the interior of an astrophysical object for cosmological distances. Indeed, they are a complementary probe with respect to other messengers such as multi-wavelength light and charged cosmic rays, allowing the observation of the far Universe and providing information on the production mechanism. Here, the neutrino telescopes in the Mediterranean Sea that are operating or in progress will be reviewed. The ANTARES (Astronomy with a Neutrino Telescope and Abyss environmental RESearch) detector is the largest neutrino telescope currently in operation in the Mediterranean Sea and the first operating in sea water. Some of the ANTARES results will be summarized, including diffuse, point-like, and multi-messenger source searches. Finally, the future km$^3$-scale telescope KM3NeT (Cubic Kilometre Neutrino Telescope) will be described focusing on the expected performances and sensitivities.

**Keywords:** neutrino; astrophysics; mass hierarchy

## 1. Introduction

Cosmic ray neutrinos are a complementary probe with respect to other messengers such as multi-wavelength light and charged cosmic rays (CR), allowing thus the observation of the far Universe and the interior of astrophysical sources.

A milestone has been set with the first evidence of a cosmic signal of high-energy neutrinos [1] by the IceCube detector [2,3]. This observation was obtained looking at a special class of events, the so-called High Energy Starting Events (HESE): being the best-fit of the cosmic neutrino flux spectral index $\alpha = 2.92^{+0.33}_{-0.29}$ [4]. However, an analysis focused on only muon neutrino events from the Northern Hemisphere showed some tensions with the measurement coming from the all-sky analysis; the best-fit of the spectral index is $\alpha = 2.19 \pm 0.10$. Several components of the extra-terrestrial neutrino fluxes, like point-like or extended galactic sources, could explain the observed discrepancy [5].

The ANTARES (Astronomy with a Neutrino Telescope and Abyss environmental RESearch) telescope [6], although much smaller than the IceCube detector, is the largest undersea neutrino telescope currently in operation. ANTARES, thanks to its exceptional visibility of the Galactic Plane and very good angular resolution [7,8], has already set important confirmations of the cosmic neutrino flux measured by IceCube [9]. The future KM3NeT (Cubic Kilometre Neutrino Telescope)—ARCA (Astroparticle Research with Cosmics in the Abyss) [10] telescope will be the ideal instrument to confirm the cosmic neutrino flux detected by IceCube. It will be a kilometer-scale detector that will have the same high visibility towards the Galactic Centre as ANTARES and also an even better angular resolution. Another detector of the KM3NeT experiment, called KM3NeT-ORCA (Oscillation

Research with Cosmics in the Abyss) [10], will be devoted to neutrino mass hierarchy studies and other fundamental neutrino physics topics.

## 2. The ANTARES and KM3NeT Detectors

The idea of a Cherenkov neutrino telescope based on the detection of the secondary particles produced in neutrino interactions was proposed in the 1960s by Markov [11]. He suggested a matrix of light detectors inside a transparent medium, like deep water or ice, for several reasons:

- A large volume of water or ice can be found easily, and it provides a large target volume for neutrino interaction.
- The deepness provides a good shielding against the secondary atmospheric particles produced by cosmic rays.
- Water and ice allow the transmission of Cherenkov light.

Cherenkov light is emitted after the passage of charged relativistic particles produced by the neutrino interaction. Neutrinos are not deflected by magnetic fields on their way towards the Earth, so their direction is pointing to the origin. Muons produced in the interactions have a kinematic angle smaller than the detector angular resolution for a high energy neutrino, so they preserve the neutrino direction. This justifies the name telescope applied to this kind of detector. Neutrinos are a unique probe to investigate the inside of astrophysical objects. Photons and protons cannot cross a dense medium, and moreover, protons are deflected by magnetic fields, so it is not possible to trace back their origin. In the case of neutrino detection due to the Charge Current (CC) $\nu_\mu$ interactions, the produced muon is the charged lepton with the longest range; it can be detected also if it is created outside the instrumented volume, and the detected event will be "track-like". The remaining CC interactions and the Neutral Current (NC) $\nu_\mu$ interaction are more difficult to detect due to the shorter range of the consequent leptons. The electron produces an electromagnetic shower, which propagates for a few meters, while the $\tau$-lepton travels some distance (depending on its energy) before it decays and produces a second shower, so those events are called "shower-like". The Cherenkov light emitted by the charged particles in the shower can be detected only if the interaction occurs inside or close to the instrumented volume of the detector.

ANTARES is the first neutrino telescope that operates in sea water. It is installed 40 km offshore from Toulon, France (42°48′ N, 6°10′ E) at a depth of 2475 m. It consists of a 3D array of 885 detection units, called Optical Modules (OMs), arranged in 12 detection lines to instrument a 0.1-km³ volume. Each line is equipped with 25 stories and three Photo-Multiplier Tubes (PMTs) in each story to detect the light produced by muons traversing the instrumented volume (each PMT is housed in an OM). The detector was completed between March 2006 and May 2008.

The KM3NeT detectors are currently under construction in the Mediterranean deep sea. Three building blocks are foreseen in the second phase of the detector construction: two KM3NeT-ARCA building blocks and one KM3NeT-ORCA block to be installed offshore from Sicily (Italy) and Toulon (France), respectively. KM3NeT's base detection unit is the Digital Optical Module (DOM). It comprises a pressure-resistant glass sphere containing 31 outward-looking PMTs. Groups of 18 DOMs are organized in lines, called detection units. Each building block counts 115 detection units. For the KM3NeT-ARCA configuration, the average horizontal spacing between detection strings is about 90 m, and the distance between two adjacent DOMs in the same string is 36 m, while for the KM3NeT-ORCA configuration, the string distance is about 23 m, and DOMs are spaced 9 m apart in the strings.

Since these DOMs contain multiple PMTs, several PMTs in one DOM can detect light from a particle, allowing a better rejection of the optical and intrinsic PMT background. The angular resolution of track-like events in KM3NeT-ARCA is below 0.1° for energies above 100 TeV, while for ANTARES, the angular resolution is below 0.4° for energy above 10 TeV. The pointing performance of the ANTARES detector has been recently confirmed by a study of the Moon shadow effect [8], and the analysis confirmed the expected angular resolution of the telescope and its pointing accuracy. The

future KM3NeT telescope will have a better angular resolution with respect to ANTARES, also in the shower-like channel, the values being 2° and 0.3°, respectively.

## 3. Latest ANTARES Results

The detection of a high energy neutrino diffuse flux by the IceCube detector has set a milestone for modern astrophysics. This discovery motivated the ANTARES search for an independent confirmation of the cosmic neutrino diffuse flux. The analysis exploited the ANTARES data sample from 2007–2015, performing an all-sky and all-flavor neutrino search [9]. Both track and shower channels have been considered, and only events coming from below the horizon were selected in order to overcome the large background of down-going atmospheric muons. The energy distribution of the reconstructed events and the expected background are reported below for both track and shower channels (Figure 1).

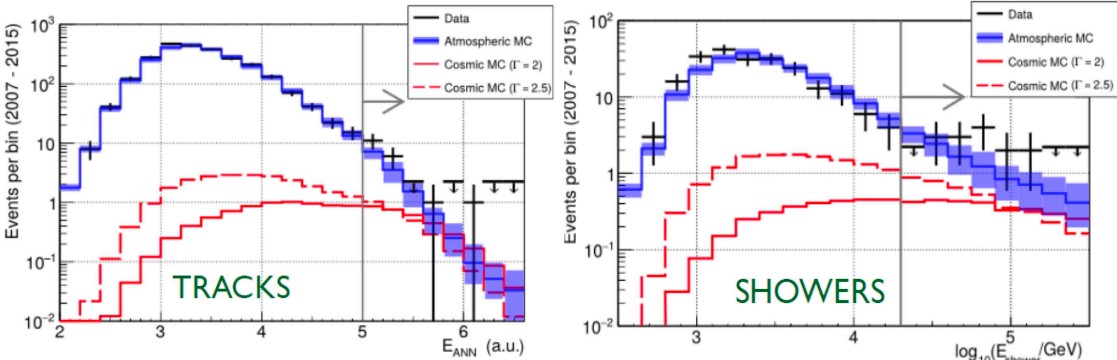

**Figure 1.** Event energy distributions of reconstructed events (black), Monte Carlo (MC) simulation of the expected atmospheric muon flux (blue), MC simulation of the expected cosmic neutrino flux (red). **Left**: track events. **Right**: shower events.

After a study of the optimized selection cuts on the event quality parameters and energy, the final sample of selected events is used to evaluate the significance of a cosmic neutrino diffuse flux (Table 1), about seven events being expected from the astrophysical flux [9].

**Table 1.** Astronomy with a Neutrino Telescope and Abyss environmental RESearch (ANTARES) diffuse flux analysis: number of selected events after quality cuts for track and shower channel. Expected number of background and signal events are also reported.

|         | Background Expectation | Number of Measured Events |
|---------|------------------------|---------------------------|
| Tracks  | $13.5 \pm 4$           | 19                        |
| Showers | $10.5 \pm 4$           | 14                        |

The ANTARES results are compatible with the IceCube cosmic neutrino flux detection, and the hypothesis of a null cosmic neutrino contribution is rejected at the 85% confidence level.

Most of the galactic gamma ray sources are in the southern sky, so ANTARES, which is located in the Northern Hemisphere, has an excellent visibility of this region of the sky. For this reason, ANTARES, being much smaller than IceCube, has a similar sensitivity for point sources in the southern sky. In Figure 2, the latest point source search results of ANTARES are reported.

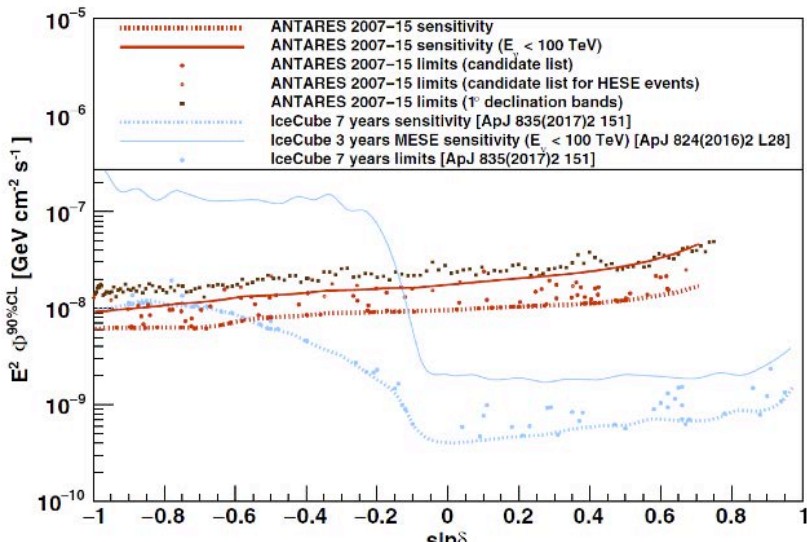

**Figure 2.** ANTARES (red) and IceCube (blue) sensitivities for point-like neutrino sources as a function of the declination of the source.

ANTARES has the best sensitivity for a large fraction of the southern sky especially for sources with energy below 100 TeV. The most significative point-like source found in the latest ANTARES analysis showed a post-trial significance of $1.9\sigma$ [12].

Thanks to its location, the ANTARES detector has also very good visibility of our galaxy. The ANTARES and IceCube collaborations recently performed a combined search for a neutrino diffuse flux from the Galactic Plane [13,14]. The combined upper limit on the neutrino flux is shown in Figure 3. The ANTARES contribution to the analysis was dominant at negative declinations, especially at neutrino cut-off energies.

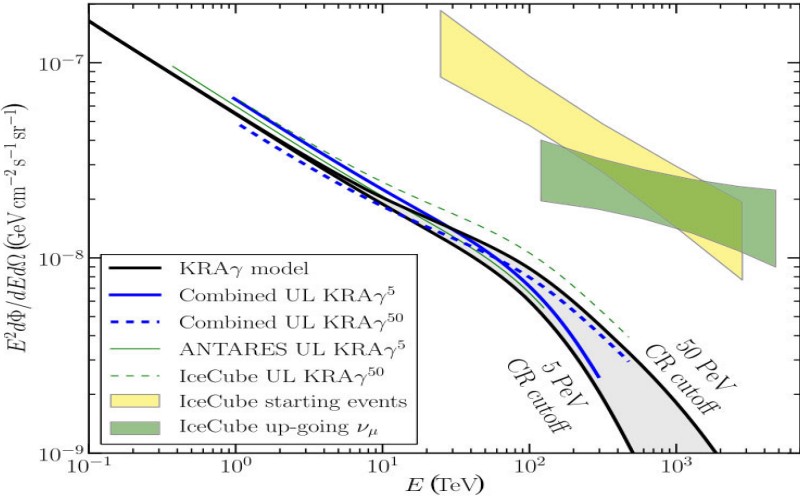

**Figure 3.** ANTARES and IceCube combined upper limit (UL) on a diffuse neutrino flux from the galactic plane (blue) and expected neutrino flux according to the KRA$\gamma$ model [15] (black). The boxes represent the isotropic astrophysical neutrino fluxes measured by IceCube using starting events (yellow) and up-going tracks (green).

The ANTARES collaboration has a wide multi-messenger program, which implies the sharing of information with various other astronomy and astrophysics experiments, ranging from electromagnetic observatories (optical, radio, X-ray, gamma-rays) to cosmic ray and gravitational wave detectors.

Transient phenomena are very interesting for a potential neutrino detection, since these analyses are almost background free in a well-defined space-time window.

The LIGO-VIRGO detection of gravitational waves (GW) was the breakthrough discovery of the last few years. A neutrino flux associated with these events could be expected [16], so the ANTARES collaboration performed several analyses in collaboration with IceCube and other cosmic ray and gamma ray detectors. Five events have been analyzed (GW150914, GW151226, LVT (Ligo-Virgo Trigger) 151012, GW170104, and GW170817), and no coincidences with neutrinos from the region of interest have been found. The emission of neutrinos is not likely in the case of black hole mergers, while neutron star mergers are more promising [17,18]. Unfortunately, the jet of GW170817, the only neutron star merger detected so far, was not aligned with our line of sight to provide a visible neutrino signal [19].

The ANTARES multi-messenger program includes also searches for neutrino fluxes from Gamma Ray Bursts (GRBs). They are the most energetic phenomenon of the Universe, and they release energies between $10^{51}$ and $10^{54}$ ergs in a few seconds. In the last years, two emission models have been proposed: the internal shock [20] and photospheric model [21]. Both models assume that the gamma ray emission is due to a relativistic jet of particles ejected by an inner engine, but the location of the interaction is different. In the case of the internal shock model, the gamma rays are produced by the interaction of different shock waves inside the jet (internal shock) and by the interaction of the jet with the interstellar medium (external shock). On the other hand, in the photospheric scenario, the interaction takes place in the initial part of the expansion of the jet, when it is still opaque to photons. The two different GRB emission models have been studied by the ANTARES collaboration, and the results are summarized in Figure 4.

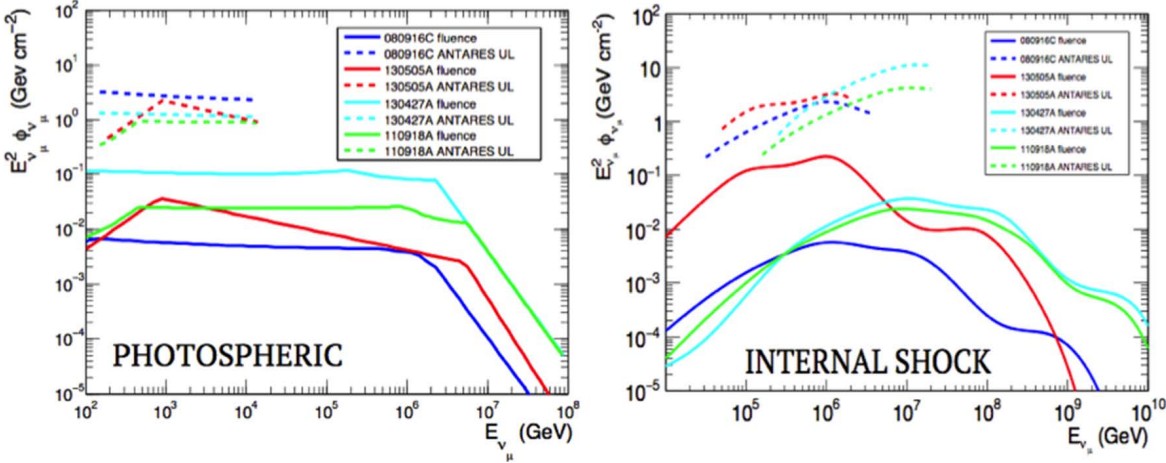

**Figure 4.** Solid lines: expected $\nu_\mu + \overline{\nu}_\mu$ fluences. Dashed lines: ANTARES 90% C.L. upper limits on selected very bright GRBs, in the energy band where 90% of the signal is expected to be detected by ANTARES. **Left**: internal shock model prediction. **Right**: photospheric model prediction.

For this search, the four brightest GRBs have been selected, and a dedicated optimization of the event selection criteria has been used. No neutrinos have been found in coincidence with the four bright gamma ray bursts, and upper limits have been derived [22].

## 4. Latest KM3NeT Results

KM3NeT will be the future km$^3$-scale neutrino telescope in the Mediterranean Sea. The construction is on-going: three strings have been deployed at the KM3NeT-ARCA site and one string at the KM3NeT-ORCA site. The deployment of several strings is expected in the last months of 2019.

The future performance of the telescope has been estimated using Monte Carlo simulation. One of the main goal of KM3NeT-ARCA will be the confirmation of the cosmic neutrino flux discovered by

IceCube. A 5$\sigma$ discovery on the diffuse IceCube flux is expected in less than a year of operation. The expected rate per year of signal neutrinos is six for track events and 16 for shower events with an expected background of four and nine, respectively [10]. The expected significance as a function of the observation time is reported in Figure 5.

KM3NET results will allow interesting studies on the IceCube signal due to its complementarity in the field of view, energy range, and flavor coverage; its location in the Northern Hemisphere allows the study of muon neutrino events from the southern sky that are not detectable by IceCube (excluding HESE). KM3NeT-ARCA will also investigate the emission of neutrino flux from point-like sources [10]. As an example, in Figure 6 (left), the expected significance as a function of the observation time for two candidate sources VelaXand RXJ1713.7-3946 is shown. KM3NeT will have a significant discovery potential for extragalactic sources, complementing the IceCube field of view, as reported in Figure 6 (right).

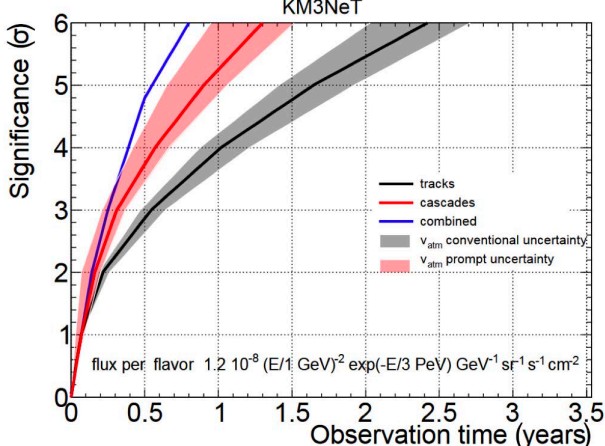

**Figure 5.** Significance as a function of the Cubic Kilometre Neutrino Telescope (KM3NeT)-ARCA (two building blocks) observation time for the detection of a diffuse flux of neutrinos corresponding to the signal reported by IceCube for the cascade channel (red line) and muon channel (black line). The black and red bands represent the uncertainties due to the conventional and prompt components of the neutrino atmospheric flux. The blue line represents the results of the combined analysis.

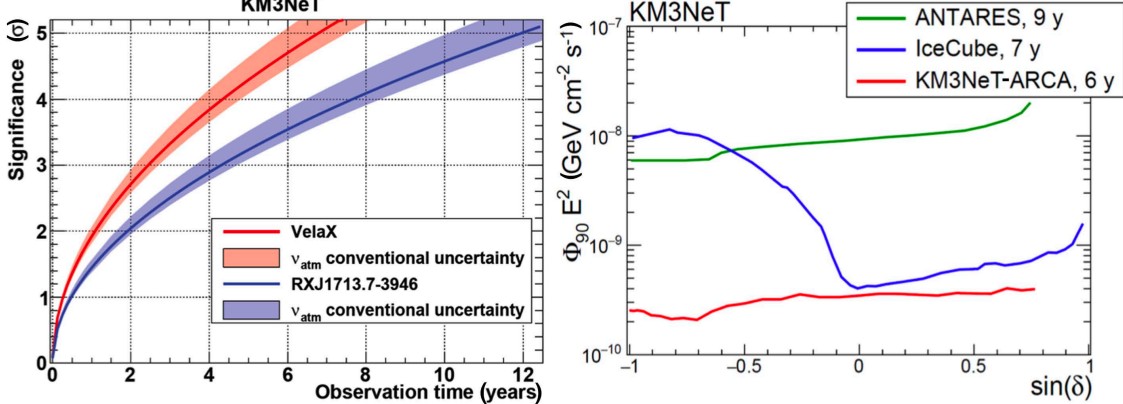

**Figure 6. Left**: Significance as a function of KM3NeT-ARCA (two building blocks) observation time for the detection of the galactic sources RXJ1713 and Vela-X. The bands represent the effect of the uncertainties on the conventional component of the atmospheric neutrino flux. **Right**: KM3NeT-ARCA (two building blocks) 5-s discovery potential as a function of the source declination (red line) for one neutrino flavor, for point-like sources with a spectrum $E^{-2}$ and three years of data taking. For comparison, the corresponding discovery potential for the IceCube and ANTARES detector (blue and green line, respectively) is shown.

One of the main goals of KM3NeT-ORCA will be the determination of the Neutrino Mass Hierarchy (NMH). The atmospheric electron neutrino flux can undergo coherent elastic forward scattering with the electrons in matter, introducing a dependence on the sign of $\Delta m_{23}^2$ [23]. The dependence on matter interaction is inverted for neutrinos and anti-neutrinos, but an asymmetry is still detectable due to the difference in the neutrino and anti-neutrino charged-current cross-sections and slightly difference in atmospheric neutrino fluxes. This property is exploited by the KM3NeT detector to determine the neutrino mass hierarchy, studying atmospheric neutrinos over a range of energies and baselines. In Figure 7 the signed $\chi^2 = (N_{NO} - N_{IO})|N_{NO} - N_{IO}|/N_{NO}$ is shown as a function of reconstructed neutrino energy and cosine zenith for track-like events and cascade-like events, where $N_{NO}$ and $N_{IO}$ are the number of reconstructed neutrinos in the hypothesis of normal ordering and inverted ordering, respectively. An overall larger statistics-only sensitivity is observed for cascades [24].

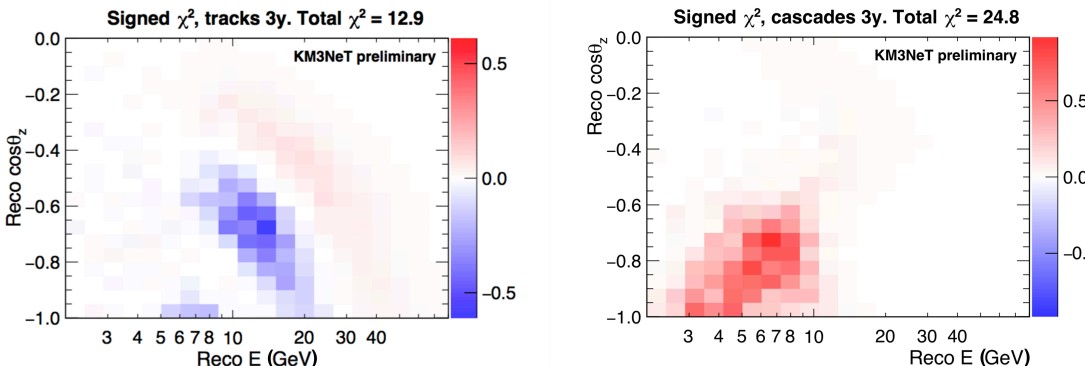

**Figure 7.** Signed $\chi^2$ as a function of the reconstructed (RECO) neutrino energy and cosine zenith for track-like events (**left**) and cascade-like events (**right**).

The neutrino mass hierarchy sensitivity is estimated with a full detector simulation; this includes atmospheric neutrinos with the flux calculated by the Honda group, atmospheric muon background, the trigger algorithms, optical water properties, and PMT response. The NMH sensitivity is calculated using a likelihood ratio approach. Figure 8 shows the KM3NeT-ORCA sensitivity to the neutrino mass hierarchy after three years of data taking [10].

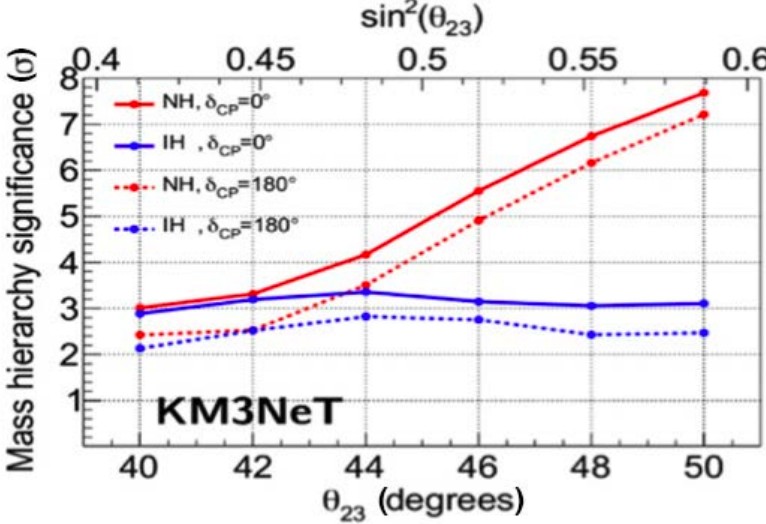

**Figure 8.** The KM3NeT-ORCA sensitivity to the neutrino mass hierarchy after three years of data taking with a full detector.

## 5. Conclusions

The ANTARES detector proved the feasibility of neutrino telescope construction and long-term operation in the deep sea. The good pointing performance of the detector and the optical properties of seawater make ANTARES ideal for the study of point-like neutrino sources. ANTARES has been able to also study a wide range of sources, from diffuse emission to gravitational wave events. KM3NeT will open a new era of neutrino astronomy. The unique design of KM3NeT multi-PMT optical modules is expected to allow a very high resolution for neutrino interaction events. The KM3NeT detector is expected to obtain important results both in neutrino astronomy and in fundamental neutrino property studies.

**Conflicts of Interest:** The authors declare no conflict of interest.

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
