# Peer review of "ANTARES and KM3NeT: The Latest Results of the Neutrino Telescopes in the Mediterranean"

_universe, doi:10.3390/universe5020065_

Reviewer 1 Report

The report is in the attachment

Author Response

Author's Notes in the attched file.

Reviewer 2 Report

This paper needs ANTARES/KM3NeT collaboration review before resubmission to any journal.

Lines 21-24: These arguments are not sound.

Lines 26-28: “ANTARES, thanks to its exceptional visibility of the Galactic Plane and very good angular resolution, hs set already important constraints on the origin of the cosmic neutrino IceCube flux.” Please provide references to support this argument.

Lines 44-45: “so it is possible to trace the muon back to the neutrino origin, since the muon maintains the same direction of the high energy neutrino.” Problematic argument. For one thing, you didn’t explain why a neutrino suddenly changes into a muon; secondly, the muon does not maintain in the exact same direction of the neutrino, the kinematic angle between the outgoing muon and the primary neutrino is energy dependent.

Syntax:

Line 3: “being absorbed and detected” to “being absorbed or detected”

Lines 7-8: “The ANTARES detector is the largest neutrino telescope currently in operation in the Northern Hemisphere and the first operating in sea water.” Is this statement true as of now? I think the GVD detector at Lake Baikal has achieved an instrumented volume larger than ANTARES as of this October. Better check again…

Line 27: typo, “hs set” to “has set”

Line 28: “the cosmic neutrino IceCube flux” to “the cosmic neutrino flux measured by IceCube”

Line 40: “CRs” to “cosmic rays”

Author Response

Author's Notes in the attached file

Reviewer 3 Report

I would like to know a little more about how “signal expectation” is calculated in table 1. Why is the assumption made of equal numbers of cascades and tracks? What is involved in mapping this number from IceCube to ANTARES and approximately what is the uncertainty in that mapping?

The text on some figures, e.g. 2 and 3, is too small to read, please increase its size.

The claim is made that neutron star mergers are promising multi-messenger candidates for neutrino telescopes.  A reference should be provided here, or otherwise some motivation for how they could produce such high-energy neutrinos as ANTARES would see.

While complementary with IceCube’s sky coverage is obvious, a further statement is made about KM3NET’s complementary to IceCube in flavor coverage. I do not know what the author means, perhaps this point could be embellished a little.

Is Fig 7 in true neutrino quantities, or reconstructed ones? Please be clear about this in the caption / on the figure.

The technical information about the likelihood construction around line 159 appears unnecessary in the context of this diverse review.

About the concluding line of the paper, about unprecedented sensitivity -  for neutrino astronomy this is clear, because of exposure to Southern sky if nothing else.  But, will the sensitivity to neutrino properties truly be unprecedented?  The example given is the mass ordering, and it is not obvious that after 3 full years with a complete detector there will not be more precise results from other programs. To reinforce this claim, some comparison should be given against the expected sensitivity of other programs at this time. Otherwise perhaps tweak the closing line a little.

Finally, some typographical / English language errors are present throughout, and I recommend a thorough proof-read by a native English speaker if possible.

Author Response

Author's Notes in the attached file

Round  2

Reviewer 1 Report

Still there is the problem with Fig. 7: both panels are identical and for

tracks.

Author Response

The figure has been been updated.